# Quantitative FRET (qFRET) Technology for the Determination of Protein–Protein Interaction Affinity in Solution

**DOI:** 10.3390/molecules26216339

**Published:** 2021-10-20

**Authors:** Jiayu Liao, Vipul Madahar, Runrui Dang, Ling Jiang

**Affiliations:** 1Department of Bioengineering, Bourns College of Engineering, University of California at Riverside, 900 University Avenue, Riverside, CA 92521, USA; vmada001@ucr.edu (V.M.); rdang018@ucr.edu (R.D.); 2Biomedical Science, School of Medicine, University of California at Riverside, 900 University Avenue, Riverside, CA 92521, USA; 3Institute for Integrative Genome Biology, University of California at Riverside, 900 University Avenue, Riverside, CA 92521, USA; 4Department of Biochemistry and Molecular Biology, Heilongjiang University of Chinese Medicine, 24 Heping Road, Harbin 150040, China; lingjneau@163.com

**Keywords:** protein interaction affinity, quantitative FRET assay, *K_D_* determination, FRET excitation, FRET quenching

## Abstract

Protein–protein interactions play pivotal roles in life, and the protein interaction affinity confers specific protein interaction events in physiology or pathology. Förster resonance energy transfer (FRET) has been widely used in biological and biomedical research to detect molecular interactions in vitro and in vivo. The FRET assay provides very high sensitivity and efficiency. Several attempts have been made to develop the FRET assay into a quantitative measurement for protein–protein interaction affinity in the past. However, the progress has been slow due to complicated procedures or because of challenges in differentiating the FRET signal from other direct emission signals from donor and receptor. This review focuses on recent developments of the quantitative FRET analysis and its application in the determination of protein–protein interaction affinity (*K_D_*), either through FRET acceptor emission or donor quenching methods. This paper mainly reviews novel theatrical developments and experimental procedures rather than specific experimental results. The FRET-based approach for protein interaction affinity determination provides several advantages, including high sensitivity, high accuracy, low cost, and high-throughput assay. The FRET-based methodology holds excellent potential for those difficult-to-be expressed proteins and for protein interactions in living cells.

## 1. Approaches to Determine Protein–Protein Interaction Affinity (*K_D_*)

Protein–protein interaction plays a central role in all the physiological activities and disease developments of living organisms, from ligand–receptor interactions, signal transductions, and cell cycles, to cancers and neurodegenerative diseases [1,2]. Due to the critical roles of protein–protein interactions and affinity alterations in daily life and in disease conditions, many technologies have been developed to determine protein interaction affinity. These techniques include—but are not limited to—surface plasmon resonance (SPR), nuclear magnetic resonance (NMR), calorimetric methods (for example, ITC—isothermal titration calorimetry and DSC—differential scanning calorimetry), radioactive labeling binding assay, ultracentrifugation, and fluorescence polarization (FP) [3,4,5]. These methods have greatly expanded our understanding of protein functions and dynamics, in basic science and in pharmaceutical development. However, most of these methods usually require tedious procedures or expensive instruments, and many of them cannot always generate reliable and consistent results due to intrinsic issues.

Most importantly, current approaches all require at least one protein partner to be purified, and a relatively large amount of protein is needed in most approaches. Due to those technique challenges, the affinities and kinetics of the difficult-to-be expressed proteins and large-scale proteomics networks are still largely unknown. The consequent functional capabilities of those proteins in physiology and pathology remain poorly understood, resulting in few therapeutics that target protein–protein interactions [4,6,7,8,9,10,11].

The Surface plasmon resonance (SPR) biosensor is a quantitative spectroscopic technique commonly used to measure binding events, specializing in the characterization of non-covalent interactions [12,13,14,15]. The method detects the binding events at a functionalized surface by monitoring the modulation of the SPR angle, caused by changes in the refractive index at the functional surface. The sensing surface is a metallic (gold, silver, or aluminum) surface interfaced with a prism. The metal surface is functionalized for the attachment of a binding partner of interest. Monochromatic light is reflected from the functionalized metal surface at an angle at which the excited metal surface electrons oscillate and generate plasmon, this is called the SPR angle. The binding partner of interest is bound to the functionalized metal surface, and potential binding molecules are flown over the functionalized surface. In the event of biomolecules binding to the bound proteins, the refractive index would change, and the SPR angle would shift. This modulation is recorded in the change of the angle and/or the intensity of the light reflected [13,16]. The advantage of this method is that it can detect both on and off rates at high sensitivity; however, the SPR method has some disadvantages. Proteins immobilized on the sensor surface may not be in their native conformation, and the heterogenicity of binding—the different orientation of ligands immobilized on the surface—may generate decreased affinity [17,18]. Moreover, due to the surface immobilization of ligands, the local concentration is higher than that of the solution, and the binding kinetic is different from ideal pseudo-first-order binding due to the mass transfer effect [18,19]. In addition, the rebinding effect and nonspecific binding to the sensor chip may occur; thus, more careful mathematic algorithms are needed to obtain meaningful parameters [20]. This SPR method for *K_D_* determination might not be valid when a simple Langmuir-type binding model does not apply [19]. If some enzymes are involved in the final product formation, leading to the interaction, inconsistent results may occur as enzymes may be lost activities during this period.

In addition to SPR, isothermal titration calorimetry (ITC), radioactive labeling, and ultracentrifugation are also used for *K_D_* determination [3,4]. These methods offer experimental convenience but also have some disadvantages. They often require environmentally unfriendly labeling or expensive instrumentation and cannot determine impure protein interactions. ITC directly quantifies heat changes that are caused by the endothermic or exothermic reaction between two interacting molecules [21,22]. ITC requires relatively large amounts (i.e., micromolar range) of highly purified proteins. Systematic errors in cell volume, heart calibration, and other errors (e.g., baseline and gas bubbles), can lead to inaccuracies [23,24,25]. It also requires relatively expensive dedicated equipment. In the ultracentrifugation assay, proteins can be nonspecifically adsorbed on the test tube walls during high-speed centrifugation. The elongated centrifugation can perturb the equilibrium between bound and free proteins, especially if the dissociation rates are fast; thus, the *K_D_* values that are determined may not represent true equilibrium constants. In principle, the fluorescent polarization (FP) approach can potentially address these issues [26]. It has been successfully used to determine protein interaction dissociation constants and has been developed into an HTS platform for small molecule drug screening. However, the FP approach requires one interactive partner to be a small molecule so that the rotation of fluorophore attached to the molecule can be significant. The FP approach also needs purified interacting partners and is less sensitive when measuring large proteins [27]. It also typically requires very sensitive instruments for good quantification, such as fluorescent microscopy, preventing its high-throughput application.

Previous quantitative FRET imaging and biochemical assay approaches have used quantitative three-cube analysis for FRET signal analysis, or ratiometric fluorescence signals to obtain FRET signals; in theory, these approaches could be used to determine protein interaction affinity (*K_D_*) in a mixture [28]. However, the quantitative three-cube approach requires measurements of molar extinction coefficients of fluorophores, as well as FRET efficiency. It also requires estimates of FRET efficiency and many instrument-dependent parameters during measurements, making it difficult to turn the approach into a general methodology [29,30]. The FRET analysis approach uses a point-to-point subtraction method to measure the FRET signal and the fluorescent ratio of acceptor and donor wavelength emissions, which does not exclude signals of direct emissions of donor and acceptor [31,32]. The application of a titration ratiometric FRET assay to determine protein interaction affinity usually yields *K_D_* values higher than those determined by SPR or ITC. The lack of accuracy and robustness of these approaches for *K_D_* determinations in a mixture is due to the occurrence of multiple fluorescence parameter estimations and absolute FRET signal determination difficulty.

## 2. Theoretical Principles of Protein–Protein Interaction Affinity (*K_D_*) Determination from Acceptor Emission Using Quantitative FRET (qFRET) Analysis

A recent quantitative FRET (qFRET) approach determines the absolute FRET signal from the subtraction of the total fluorescence signal with the fluorescent signals of the free donor and the free acceptor—as obtained from the donor or acceptor emissions—multiplied, respectively, by their cross-wavelength correlation constants [33,34]. The *K_D_* values determined with this approach, which we refer to here as qFRET, are in excellent agreement with the values determined using SPR and ITC; this agreement demonstrates its accuracy. In addition, this approach was carried out in a multi-well plate, and therefore it can be expanded to the high-throughput mode. 

According to the mass action law of molecular interaction, to determine the protein interaction equilibrium constant (*K_D_*) an essential step is to quantify the absolute FRET signal resulting from the interactive protein complex, to elucidate the *K_D_* from the FRET signal [33,35] (Figure 1). In the past, several efforts were made to elucidate the absolute FRET signal from the fluorescence emission at the acceptor emission wavelength, including the three-cube method [28,36]. However, these methods need to determine the quantum yield of fluorophores, fluorophore distance, and FRET efficiency, which heavily depends on instruments and assumptions. Therefore, they could not determine FRET signals consistently and were limited to the microscope image approach.

We developed a FRET signal analysis method to differentiate FRET signals from the emissions of free ligands and acceptors. In this approach to determine protein interaction *K_D_* from the FRET signal, the first step is to obtain the absolute FRET signal (Em_FRET_) from the interactive protein complex, to determine the maximum bound protein partner. The direct emissions of the donor (CyPet) and acceptor (YPet) need to be determined. They are excluded from the total emission at the FRET emission signal wavelength (530 nm in the FRET pair, CyPet and YPet). However, the FRET emission signal wavelength at 530 nm consists of three components: the direct emission of the donor, the direct emission of the acceptor (which does not come from the protein interaction), and the emission of the FRET signal (Em_FRET_), which comes from the protein interaction (Figure 2A). In order to obtain the EmFRET, the direct emissions of donor and acceptor need to be determined. In order to measure the direct emissions of the donor and acceptor in the FRET assay, a cross-wavelength co-efficiency approach was developed. First, the mixture of donor and acceptor is excited at the donor excitation wavelength (414 nm), and two emission signals at 475 nm (FL_DD_) and 530 nm (FL_DA_) are determined, respectively. The fluorescence emissions at 475 nm, when excited at 414 nm, are from both the emission of the donor (FL_DD_) and the direct emission of acceptor at 475 nm, which is very small as compared with the donor emission (<2.6% of emission in the case of CyPet and YPet) and can be neglected. The direct emission of the donor at FRET wavelength (530 nm) is proportional to its emission at 475 nm, when excited at 414 nm with a constant ratio factor of *a*, which is defined as FL_DA_/FL_DD_ (Figure 2B). The *a* factor can be predetermined using pure donor fluorescence protein. Therefore, the emission of the donor at the FRET emission wavelength (530 nm) can be calculated as *α* * FL_DD_. On the other hand, the direct emission of the acceptor at the FRET emission wavelength (530 nm), excited at the donor excitation wavelength (414 nm) (FL_AD_), is proportional to its emission at the FRET emission wavelength (530 nm), excited at 495 nm (FL_AA_), with a constant ratio factor *β*, which is defined as FL_AD_/FL_AA_ (Figure 2C). The FL_AA_ is the fluorescence emission of the acceptor at 530 nm when excited at the acceptor excitation wavelength (495 nm).

After the emissions of the FRET donor at 475 nm—excited at 414 nm and the FRET acceptor at 530 nm—excited at 495 nm—are determined, the absolute FRET emission signal can be calculated by subtracting the total emission at 530 nm—excited at 414 nm with the above two signals with multiplications of the ratios of *α* and *β*, respectively, from Figure 3A. In other words, the FRET emission signal (Em_FRET_) can be determined using the following Equation (1):E_mFRET_ = FL_DA_ − *α* * FL_DD_ − *β* * FL_AA_(1)
where the ratio constants *α* and *β* are first experimentally determined.

Once we obtain E_mFRET_, following the general mass action law as (2):Protein 1-FRET donor + Protein 2-FRET acceptor ⟷ Protein 1-FRET donor · Protein 2-FRET acceptor(2)
where Protein 1 is tagged to the FRET donor (Protein 1-FRET donor) and Protein 2 is tagged to FRET acceptor (Peotein 2-FRET acceptor).

The equilibrium constant, *K_D_*, can be defined as in Equation (3) or Figure 1C.
(3)$$K_{D}=\frac{{\left[\mathrm{Protein}\;1-\mathrm{Donor}\right]}_{\mathrm{free}}{\left[\mathrm{Protein}\;2-\mathrm{Acceptor}\right]}_{\mathrm{free}}}{\left[\mathrm{Protein}\;1-\mathrm{Donor}\;\cdot \;\mathrm{Protein}\;2-\mathrm{Acceptor}\right]}=\frac{{\left[\mathrm{CyPetRanGAP}1c\right]}_{\mathrm{free}}{\left[\mathrm{YPetUbc}9\right]}_{\mathrm{free}}}{{\left[\mathrm{YPetUbc}9\right]}_{\mathrm{bound}}}$$

This can be rearranged to
(4)$$\left[\mathrm{Protein}\;1-\mathrm{Donor}\cdot \mathrm{Protein}\;2-\mathrm{Acceptor}\right]=\frac{{\left[\mathrm{Protein}\;2-\mathrm{Acceptor}\right]}_{\mathrm{bound}\;\max}{\left[\mathrm{Protein}\;2-\mathrm{Acceptor}\right]}_{\mathrm{free}}}{K_{D}+{\left[\mathrm{Protein}\;2-\mathrm{Acceptor}\right]}_{\mathrm{free}}}$$
where [Protein 2-Acceptor]_boundmax_ is the theoretical maximal Protein 2-Acceptor concentration that is bound to Protein 1-Donor concentration, and [Protein 2-Acceptor]_free_ is free Protein 2-Acceptor concentration. [Protein 2-Acceptor]_bound_ is proportional to the FRET signal of bound proteins; then, Equation (4) can be converted into Equation (6) using the relationship shown in Equation (5):(5)$${\left[\mathrm{Protein}\;2-\mathrm{Acceptor}\right]}_{\mathrm{bound}}/{\left[\mathrm{Protein}\;2-\mathrm{Acceptor}\right]}_{\mathrm{boundmax}}\;=\;{\mathrm{Em}}_{\mathrm{FRET}}/{\mathrm{Em}}_{\mathrm{FRET}}$$
(6)$${\mathrm{Em}}_{\mathrm{FRET}}={\mathrm{Em}}_{\mathrm{FRET}\;\max}\left(1-\frac{2K_{D}}{X-A+K_{D}+\sqrt{\left(X-A-K_{D}\right)2+4K_{D}X}}\right)$$
where Em_FRET_ is the absolute FRET signal and can be determined using Equation (1). The Em_FRET max_ is the absolute FRET signal when the maximum amount of Protein 2-Acceptor is bound by Protein 1-Donor. The A in Equation (6) is the concentration of Protein 1-Donor in the binding assay and X is the various titration concentrations of Protein 2-Acceptor added into the binding reaction. After the Em_FRET_ values are determined using Equation (1) at various concentrations of Protein 2-Acceptor (Figure 3B), *K_D_* and E_mFRET max_ can be determined through the multiple variable regression in Equation (6).

The quantitative FRET assay has been carried out in the 384 well plate, and fluorescence signals were determined using fluorescence plate readers. Therefore, this approach can be implemented in a high-throughput technology platform that can be implemented for large-scale measurement. The FRET acceptor emission approach has been used for many protein–protein interaction affinity *K_D_* determinations of SUMOylation cascade and others. The *K_D_* between SUMO1 peptide and E2 conjugation enzyme Ubc9 was determined at 0.33 ± 0.04 μM, which is compatible with the value of 0.35 μM determined with the SPR, or the value of 0.25 ± 0.07 μM determined with ITC [1,2,3].

## 3. Determination of Protein–Protein Interaction Affinity, *K_D_*, from FRET Donor Quenching 

In FRET, the fluorescence quenching of a donor is proportional to the energy transferred to its acceptor, which is proportional to the concentration of the protein–protein bound complex. In contrast, fluorescence quenching is a more general approach than fluorescence emission, as many FRET acceptors are quenching fluorophores, especially in small molecule FRET pairs [37]. The fluorescence quenching approach was pioneered by Velick et al. for characterizing antibody–hapten binding [38] Liu and Schultz further developed this approach for characterizing binding between macromolecules [39].

As the FRET signal emitted by the FRET acceptor is proportional to the amount of FRET donor and acceptor bound complex, the decrease of the FRET donor signal—also called the quenching of donor—should also be proportional to the bound complex, and therefore could be used for *K_D_* determination [40].

Starting with the general law of mass action for the protein–protein interaction in above Equation (2), the emission intensity of Protein 1-FRET donor at its emission wavelength decreases (475 nm in the case of CyPet) by the acceptor Protein 2-FRET acceptor as FRET occurs (Figure 4A). Because the amount of quenched Protein 1-FRET donor is proportional to the amount of bound protein complexes, the relationship of emission decreases and the concentration of bound protein can be represented as follows:
ΔEm_475_ = n × [Peotein 2-FRET acceptor]_bound_(7)
where n is a constant related to the FRET efficiency, affinity, and distance between Protein 1-FRET donor and Protein 2-FRET acceptor; [Protein 2-FRET acceptor]_bound_ is the concentration of bound Protein 2-FRET acceptor; ΔEm_475_ is the decrease of emission intensity at the donor emission wavelength, 475 nm—when excited at its wavelength of 414 nm—at each specific concentration of Protein 2-FRET acceptor. As follows:(8)$${\Delta \mathrm{Em}}_{475}={\mathrm{Em}}_{475\left(\left[\mathrm{Protein}\;2-\mathrm{FRET}\;\mathrm{acceptor}\right]=\;X\right)}-{\mathrm{Em}}_{475\left(\left[\mathrm{Protein}\;2-\mathrm{FRET}\;\mathrm{acceptor}\right]=0\right)}$$
where x is the concentration of the FRET acceptor in the FRET assay. 

If we set the total concentration of Protein 1-FRET donor to a constant A, the concentration of total Protein 2-FRET acceptor to the variable X, and ΔEm_475_ to the variable Y, we can convert the concentration of bound and free Protein 1-FRET or Protein 2-FRET acceptor proteins in Equation (2) to:(9)$${\left[\mathrm{Protein}\;2-\mathrm{FRET}\;\mathrm{acceptor}\;\right]}_{\mathrm{bound}}=\frac{Y}{n}$$
and
(10)$${\left[\mathrm{Protein}\;1-\mathrm{FRET}\;\mathrm{donor}\right]}_{\mathrm{free}}=A-\frac{Y}{n}$$
and
(11)$${\left[\mathrm{Protein}\;2-\mathrm{FRET}\;\mathrm{acceptor}\;\right]}_{\mathrm{free}}=X-\frac{Y}{n}$$

Based on the definition of *K_D_* and Equations (9)–(11), we can derive
(12)$$K_{D}=\frac{\left(A-\frac{Y}{n}\right)\left(X-\frac{Y}{n}\right)}{\frac{Y}{n}}=\frac{\left(\mathrm{nA}-Y\right)\left(\mathrm{nX}-Y\right)}{\mathrm{nY}}$$

After rearranging the above equations, the following equation is obtained:(13)$$Y^{2}-n\left(A+X\right)Y+n^{2}\mathrm{AX}=K_{D}\mathrm{nY}$$
(14)$$Y^{2}-n\left(A+X+K_{D}\right)Y+n^{2}\mathrm{AX}=0$$
(15)$$Y=\frac{n}{2}\left(A+X+K_{D}-\sqrt{{\left(A+X+K_{D}\right)}^{2}-4\mathrm{AX}\;}\right)$$

Therefore, by fitting ΔEm_475_ (Y) and the total Protein 2-FRET acceptor concentration, X, as a multi-variable regression with Equation (15), we can derive the value of *K_D_* and the constant n. 

The method of FRET quenching for *K_D_* determination has been used in the protein–protein interaction affinity *K_D_* determination of the same protein pair of SUMO1–Ubc9 [1]. The *K_D_* value is 0.47 ± 0.03 μM, which is compatible with the value of 0.41 ± 0.02 μM from the acceptor emission approach.

## 4. Comparison of Protein Interaction Affinity (*K_D_*) Determinations through FRET Acceptor Emission and FRET Donor Quenching Methods

To directly compare the sensitivity and accuracy of the FRET acceptor emission or the FRET donor quenching methods, we took both approaches to determine the same protein–protein interactions, CyPet–SUMO1 and YPet–Ubc9 [33,35,40]. Data were taken from the experiments in four different conditions—in which CyPet–SUMO1 was set to 0.1, 0.5, 1.0, or 1.5 μM, respectively, in order to verify each other—and compared with the values of *K_D_* determined by the FRET emission method. The *K_D_* values from the acceptor FRET emission method were 0.36 ± 0.02, 0.36 ± 0.01, 0.41 ± 0.03, and 0.42 ± 0.06 μM at the same concentrations of CyPet–SUMO1, respectively. To minimize the effect of variations, we performed global optimization for the *K_D_* values from the two approaches. The *K_D_* values from the fluorescence quenching and the emission method were 0.47 ± 0.03 and 0.41 ± 0.02 μM, respectively. The statistic bootstrap analysis suggests the two *K_D_* values are indeed equal. The data were also very close to the results of the *K_D_* values (i.e., 0.35 μM by surface plasma resonance (BIACORE) [33] and 0.25 μM with isothermal titration calorimetry) [41], indicating a good agreement of the results.

## 5. Conclusions and Future Directions

In this review, we systematically summarized the novel developments of both the donor quenching-based and the acceptor emission-based methods for *K_D_* determination. The FRET-based *K_D_* determinations are the accurate and sensitive approach to determine protein–protein, or other molecular interactions, affinities. The FRET-based *K_D_* determinations can provide several advantages over other methods. First, FRET measurement for interactive proteins is carried out in a solution, and this condition mimics the physiological environment of living cells. In contrast, other methods, such as SPR, require that the conjugations take place on a chip surface, which could interfere with the interactions of proteins. Second, the FRET assay is very sensitive. The concentration of fluorescence-tagged proteins in the FRET assay can be as low as nM to pM, depending on the PMT or CCD camera detectors; therefore, no significant amount of proteins is needed to determine their *K_D_*. Third, the FRET-based *K_D_* determination method is environmentally friendly, and the protein labeling method is universal. The FRET-based *K_D_* determination can be expanded into FRET assays without protein purifications. This improvement is particularly critical for those difficult-to-be expressed or purified proteins, and for in vivo *K_D_* determination. In addition, the FRET assay is conducted in a 384 well plate format, which can be expanded to large-scale network biology kinetics in the future.

## Figures and Tables

**Figure 1 molecules-26-06339-f001:**
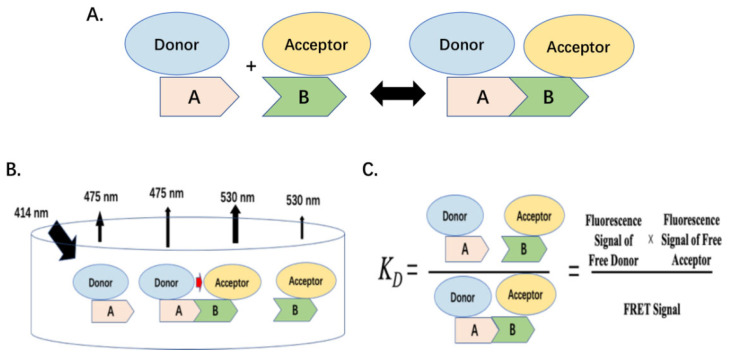
FRET-based protein equilibrium constant *K_D_* determination: (**A**). Mass action of protein-protein interaction with FRET label; (**B**). Diagram of fluorescence signals of protein-protein interaction in solution; (**C**). Schematic diagram of FRET-based protein equilibrium constant *K_D_* determination.

**Figure 2 molecules-26-06339-f002:**
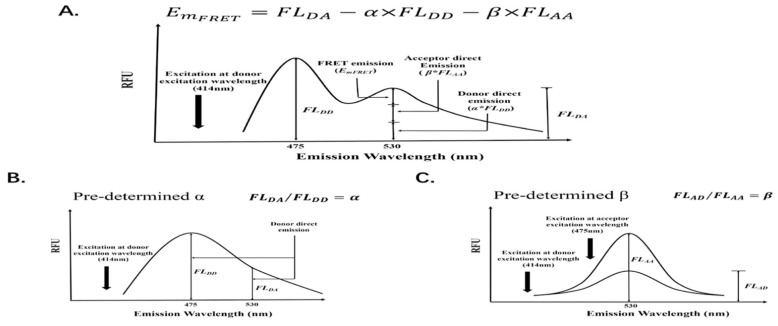
Quantitative FRET signal analysis: (**A**). Fluorescent emission of FRET donor and acceptor mixture at 530 nm, when excited at 414 nm (FL_DA_), can be divided into three components: FRET emission from donor, direct emission of donor, and direct emission of acceptor; (**B**). The cross-wavelength correlation coefficiency of donor *α* is defined as the fluorescent emission of donor at 530 nm, when excited at 414 nm (FL_DA_), divided by the fluorescent emission of donor at 475 nm, when excited at 475 nm (FL^AA^); (**C**). The cross-wavelength correlation coefficiency *β* of the acceptor is defined as the fluorescent emission of donor at 530 nm, when excited at 414 nm (FL_AD_), divided by the fluorescent emission of donor at 530 nm, when excited at 475 nm (FL_AA_).

**Figure 3 molecules-26-06339-f003:**
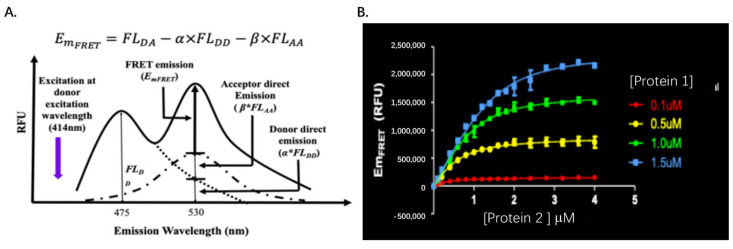
FRET acceptor emission signal for protein equilibrium constant *K_D_* determination: (**A**). The FRET acceptor emission, as resulting from protein–protein interaction (E_mFRET_), is extracted from the total fluorescence emission at the acceptor emission wavelength and is used to determine protein–protein bound complex; (**B**). E_mFRET_ signals at four concentrations of donor are determined using the method in A.

**Figure 4 molecules-26-06339-f004:**
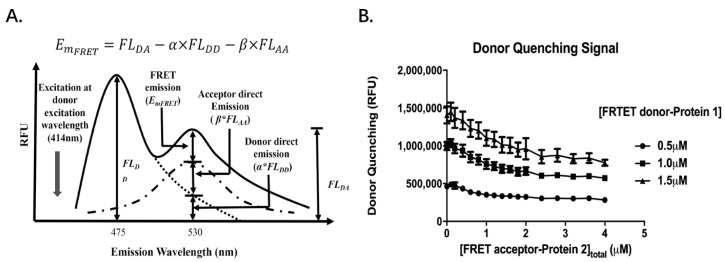
FRET donor quenching signal for protein equilibrium constant *K_D_* determination: (**A**). The FRET donor quenching resulted from protein–protein interaction is used to determine protein–protein bound complexes; (**B**). The FRET donor quenching signals at three concentrations of donor are determined with increasing concentrations of FRET acceptor, using the method in (**A**).

## Data Availability

No new data were created or analyzed in this study. Data sharing is not applicable to this article.

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
