# Peer review of "Quantitative FRET (qFRET) Technology for the Determination of Protein–Protein Interaction Affinity in Solution"

_molecules, 2021, doi:10.3390/molecules26216339_

Round 1

Reviewer 1 Report

The manuscript entitled “Quantitative FRET (qFRET) Technology for Protein-Protein Interaction Affinity Determination in Solution ” was submitted by Liao et al. for publication in Molecules. First of all the composition of this manuscript is rather unusual for scientific papers. There is no division of the article in introduction, results (and discussion), conclusion sections. Instead, the authors provide long and very basic information about SPR, ITC and FRET. I do not see any connection between this kind of “Introduction” and content of this paper. In the following 3 chapters the applicability of FRET for the analysis of protein-protein interaction is described. However, there is NO experimental examples of the applicability of their approach for these studies.

I do not recommend the publication of this article.

Author Response

I would like to thank the reviewer’s valuable comments! I believe there may be some misunderstandings about the type of manuscript. This is not a research article, but rather a review manuscript for quantitative FRET analysis and its applications in protein-protein interaction affinity determinations. Each section is very closely connected in a logical way. The first section (1) is general comparisons of three major methods of protein-protein interaction affinity determinations, including SPR. ITC and qFRET; the second section(2) is a detail description of quantitative FRET(qFRET) and protein-protein interaction affinity determination using FRET acceptor emission. This method can mainly be used in fluorescent protein FRET; the second section(3) is a detail description of protein-protein interaction affinity determination using FRET donor quenching. This method can mainly be used in small molecule FRET; The section (4) is a comparison of these two major FRET methods; The last section(5) is a conclusion of qFRET method for KD determinations and future direction. 

            Because this is a review manuscript, we did not provide any experimental examples, in original draft. Now we added experimental examples in each section.

            The English language has been revised.  

Reviewer 2 Report

The manuscript from Liao et al. reviewed the determination of protein-protein interaction affinity with quantitative FRET methodology by either measuring FRET donor quenching or acceptor emission. The qFRET method described has several benefits compared to other Kd determination methods in that it requires little amount of sample, is cost-effective, accurate, and potentially could be applied to impure protein preps and the study of protein interaction in cells. Therefore, it could appeal to many readers interested in the field. The authors logically laid out the background and presented the theory of interest,  with sufficient details and easy-to-follow illustrations. I recommend the acceptance of the article.

Here are some minor suggestions: 

1.The authors could consider adding statements addressing the novelty of the manuscript in comparison to the published papers. 

2.Line   17-18, “But due to complicated procedure or challenges in differentiate FRET signal from other direct emission signals form donor and receptor, the progress was slow.” “differentiate“ should be “differentiating”, and “form donor and receptor” should be “from donor and receptor”.  

Author Response

I would like to thank the reviewer’s encouragement and very positive comments!

Here are some minor suggestions: 

  1. The authors could consider adding statements addressing the novelty of the manuscript in comparison to the published papers. 

Yes. We added some sentences in both Introduction and Conclusion.

In this review, we will focus on recent developments of the quantitative FRET analysis and its application in protein-protein interaction affinity, KD, determination, either through FRET acceptor emission or donor quenching methods. We mainly review the novel theatrical development and experimental procedures, rather than specific experiments.”

In this review, we systematically summarize the novel developments of both either though donor quenching-based or acceptor emission-based methods for KD determinations.”

2.Line   17-18, “But due to complicated procedure or challenges in differentiate FRET signal from other direct emission signals form donor and receptor, the progress was slow.” “differentiate“ should be “differentiating”, and “form donor and receptor” should be “from donor and receptor”.  

            Thanks! They have been corrected.

Reviewer 3 Report

The manuscript of Liao et al. describes the application of quantitative FRET technology to determine the affinity of protein-protein interactions. The topic is interesting, since the study of protein-protein interactions is assuming extreme importance. Unfortunately the manuscript has important shortcomings, if identified as a review. Current approaches to protein-protein interaction affinity should be described in more detail. The second part of the manuscript looks more like a research article than a review. In fact, there are no reported previous examples of this technique present in the literature, but only results performed by the authors for a protein-protein interaction they studied (they report the affinity data obtained). A review must report the results in the literature, obtained from different research groups, to give the state of the art of the subject.

In my opinion the manuscript cannot be accepted as a review. I invite the Authors to submit a more complete and more relevant paper to a review.

Author Response

I would like to thank the reviewer’s positive opinions about the topic and valuable suggestions! In this manuscript, we did summarized the three other FRET-based KD determination methods from three other groups in the Introduction, Dr.Yue, Dr. Hay, and Dr.Linderman[29-32]. We have been followed this filed very tighly. Unfortunitely, there are not many other efforts or literatures on the topic of FRET-based KD determination over the years. The editor for the special issue particularly invited us to mainky summerize our approach. Therefore, we decided to review the two major approahces in details so that readers can easily capture the major concepts and theratical development in one review, other than reading different research papers. We also added results of KD values determined from both acceptor emission and donor quenching, as well as comparision of the two methods. In the title of the manuscript, we particular narrow down the content of the manuwcript as specific to the Quantitative FRET (qFRET) Technology for Protein-Protein Interaction Affinity Determination, which are mainly driven by our group currently. I will be more than happy to see more groups to work on this very important topic in the future.

Round 2

Reviewer 3 Report

The paper is improved and Authors responded quite convincingly to the problems requested by the referee.

It is necessary to carefully check the units of measurement of the Kd values.

For example:

Line 249, 0.33 ± 0.04 mM and 0.35 mM

Line 250, 0.25 ± 0.04 mM and so on.

Instead line 308, 0.36 ± 0.02 and 0.42 ± 0.06 M

Line 311, 0.47 ± 0.03 and 0.41 ± 0.02 without unit of concentration

Line 313, 0.35 μM

Line 314, 0.25 (symbol?)M

Provided that these issues are settled, the revised version of paper is worthy of publication.

Author Response

Dear Reviewer,

Thank reviewer very much for your quickly turn-around handling for our manuscript! I have read the comments from the reviewer 3 and revised the manuscript as following,

Review 3

The paper is improved and Authors responded quite convincingly to the problems requested by the referee.

            I would like to thank the reviewer’s positive comment for the revision!

It is necessary to carefully check the units of measurement of the Kd values.

For example:

Line 249, 0.33 ± 0.04 mM and 0.35 mM

Line 250, 0.25 ± 0.04 mM and so on.

Instead line 308, 0.36 ± 0.02 and 0.42 ± 0.06 M

Line 311, 0.47 ± 0.03 and 0.41 ± 0.02 without unit of concentration

Line 313, 0.35 μM

Line 314, 0.25 (symbol?)M

            I apologize for these errors although some of them were probably due to the formatting issue. Thank the reviewer very much for the careful readings!

Sincerely,

Jiayu Liao